# Retinal Microcirculation Changes in Crohn’s Disease Patients under Biologics, a Potential Biomarker of Severity: A Pilot Study

**DOI:** 10.3390/jpm12020230

**Published:** 2022-02-07

**Authors:** Eloi Debourdeau, Chloé Charmard, Isabelle Carriere, Julien Plat, Max Villain, Lucile Boivineau, Romain Altwegg, Vincent Daien

**Affiliations:** 1Department of Ophthalmology, Gui de Chauliac Hospital, F-34000 Montpellier, France; c-chamard@chu-montpellier.fr (C.C.); j-plat@chu-montpellier.fr (J.P.); m-villain@chu-montpellier.fr (M.V.); 2Institute for Neurosciences of Montpellier INM, University Montpellier, INSERM, F-34091 Montpellier, France; isabelle.carriere@inserm.fr; 3Department of Gastroenterology and Hepatology, Saint-Eloi Hospital, F-34000 Montpellier, France; l-boivineau@chu-montpellier.fr (L.B.); r-alwegg@chu-montpellier.fr (R.A.); 4The Save Sight Institute, Sydney Medical School, The University of Sydney, Sydney, NSW 2000, Australia

**Keywords:** OCT angiography, fundus photographs, Crohn’s disease, cardiovascular disease, population-based study, retinal vascular network, vascular predicting capacity, vascular prevention

## Abstract

Crohn’s disease (CD) is associated with increased cardiovascular risk and the retinal microcirculation is a reflection of the systemic microcirculation. Is the retinal microcirculation altered in relation to the severity of Crohn’s disease? This cross-sectional case-controlled study was conducted in a university hospital center from November 2020 to February 2021. We prospectively included patients with moderate (biologic therapy) or severe (biologic therapy + peri-anal disease and/or digestive resection) CD and age- and sex-matched controls. Individuals with diabetes, renal disease, cardiovascular disease, ophthalmological history or poor quality images were excluded. All participants underwent OCT angiography (OCT-A) imaging (Optovue, Fremont, CA). Analysis of covariance was used. 74 CD patients (33 moderate, 41 severe) and 74 controls (66 (44.6%) men; mean (SD) age 44 (14) years) were included. Compared with the controls, the severe CD patients showed a significantly reduced mean foveal avascular zone area (*p* = 0.001), superficial macular capillary plexus vessel density (*p* = 0.009) and parafoveal thickness (*p* < 0.001), with no difference in mean superficial capillary flow index (*p* = 0.06) or deep macular capillary plexus vessel density (*p* = 0.67). The mean foveal avascular zone was significantly lower in the severe than the moderate CD patients (*p* = 0.010). OCT-A can detect alterations in retinal microcirculation in patients with severe versus moderate CD and versus age- and sex-matched controls.

## 1. Introduction

Crohn’s disease (CD) is a chronic immune-mediated disease characterized by inflammatory lesions that may involve any digestive layer in any part of the entire gastrointestinal tract from the mouth to the anus [1]. It is a systemic inflammatory disease, characterized by an inappropriate inflammatory response to modified gut microbiota in patients with genetic susceptibility [2]. It is a public health issue with an increasing incidence ranging from three to twenty cases per 100,000 in North America in 2012 [3]. The diagnosis is established from a set of clinical, endoscopic, histological and/or radiological arguments [4]. Symptoms are variable and include abdominal pain, diarrhea, vomiting, weight loss and anal involvement [5]. CD is associated with extra-intestinal manifestations with, in order of frequency, arthritis (33%), aphthous stomatitis (10%), uveitis (6%), erythema nodosum (6%), axial spondyloarthropathy, psoriasis, pyoderma gangrenosum and primary sclerosing cholangitis [6]. Management of the disease is a step-up approach, with first medical treatment and then surgery in complicated or refractory cases. The therapeutic target is to achieve clinical and endoscopic remission to avoid complications and recourse to surgery [7]. Inflammatory bowel diseases (IBD) (CD and ulcerative colitis) are associated with an increased risk of acute arterial events and cardiovascular disease independent of other cardiovascular risk factors [8,9,10]. This increased cardiovascular risk is related to chronic systemic inflammation that results in endothelial dysfunction and platelet aggregation, leading to atherosclerosis [11].

The retinal microcirculation visualized by OCT-angiography (OCT-A) is very sensitive to capillary damage and a potential biomarker of the overall microcirculation in cardiac, cerebral and renal diseases [12,13,14,15], and of the cardiovascular risk profile [14]. Altered retinal microcirculation has been described in active IBD [16], and increased arteriolar tortuosity has also been reported in IBD, probably due to systemic vascular damage [17]. In the same way, altered OCT-A findings were described in rheumatoid arthritis, another chronic systemic inflammatory disease with increased cardiovascular risk [18,19], and in lupus erythematosus, which is also associated with a high thrombotic risk [20]. The increased cardiovascular and prothrombotic risk in CD could be associated with a rarefaction of the retinal microcirculation, which can be observed in OCT-A by a decrease of the microvascular density or an enlargement of the central avascular zone [21]. Hence, retinal microcirculation in CD could be altered in severe disease, and its status examined by OCT-A could be a biomarker of severity.

In the present study, we aimed to compare the retinal microcirculation of CD patients under biologic therapy with age- and sex-matched controls, according to disease severity.

## 2. Materials and Methods

### 2.1. Design

This cross-sectional case–control study was approved by the Institutional Review Board of Montpellier University Hospital (IRB ID: 202000607). Written informed consent was obtained for each eligible participant in accordance with the 1995 Declaration of Helsinki.

### 2.2. Study Participants

The present study was conducted in the department of gastroenterology and ophthalmology of Montpellier University Hospital (France). We included consecutive CD patients receiving biologic agents (anti-TNF agents, IL-12/23 inhibitors, anti-integrin agents and Janus kinase 1 inhibitors) in the department of gastroenterology and without an ophthalmologic history or symptoms.

The control participants were age-matched (±5 years) and sex-matched with the CD patients. They were healthy volunteers solicited by Montpellier University Hospital, with no ocular or systemic (e.g., digestive history, diabetes or renal disease) medical history.

The exclusion criteria for all participants were: a history of retinal disease, glaucoma, refractive error ≥±4, previous ophthalmologic surgery, diabetes, cardiovascular disease, renal disease, poor-quality images (quality index (Q-score) <6 or artefacts), age < 18 years, under guardianship and lack of health insurance.

### 2.3. Eye Examination

Automatic refraction and intraocular pressure were measured by using the Nidek Tonoref II autorefractokeratometer (Nidek Ltd., Gamagori, Japan). Keratometries were converted to the spherical equivalent power. OCT-A scans were performed with an Avanti spectral domain OCT-A with the XR Avanti AngioVue OCT-A software (Optovue, Fremont, CA, USA). A 3 mm × 3 mm macular scan (Angio Retina (3.0)) and a 4.5 mm × 4.5 mm diameter peripapillary scan (Angio Disc (4.5)) were performed after 5 min of calm sitting (Figure 1).

Perfusion vascular density (VD) and foveal avascular zone (FAZ) were automatically measured over the 3-mm × 3-mm EDTRS circle. The proprietary software segmented the OCT-A scans into the superficial (SCP) and deep capillary plexus (DCP) (Figure 2).

The SCP perfusion vessel density (SCP-VD) and DCP perfusion vessel density (DCP-VD) were measured into the whole EDTRS circle of 3 mm × 3 mm (VD whole) and into the parafoveal region (VD parafoveal), defined as a ring centered by the fovea with an inner diameter of 0.6 mm and an outer diameter of 2.5 mm. The optic disc head (ONH) VDs were obtained from the 4.5 mm × 4.5 mm diameter peripapillary scan: ONH whole image, ONH inside disc and radial peripapillary (RPC) VD.

The flow indexes (FI) were measured using a 1 mm radius disc centered by the fovea from the 3-mm × 3-mm angiogram [22]. The DCP flow index (DCP-FI), the SCP flow index (SCP-FI) and the choriocapillaris plexus flow index (CCP-FI) were assessed from the limits described in Figure 2.

The VD and the FI were respectively defined as the percentage area occupied by the vascular and as the mean decorrelation value (correlated with blood flow velocity) in the measurement regions.

The average retinal nerve fiber layer thickness (RNFL), the ganglion cell complex (GCC) and the parafoveal retinal thickness (RT) were analyzed. All measurements were performed by the first author or a trained orthoptist under supervision unaware of patient status. Only the right eye was considered, because of high correlation between eyes [23]. To test the inter-operator reproducibility (ICC), scans of 30 eyes were acquired by the 2 examiners. The ICC was excellent (Pearson’s correlation: FAZ area, r = 1.0; VD whole, r = 0.85; DCP-FI, r = 0.96; all *p* < 0.001).

### 2.4. Clinical Assessment and Data Recording

Each CD patient had a confirmed CD diagnosis, and disease activity and severity, before enrollment by an IBD specialist. A collegial decision by the gastroenterologists classified the disease as “moderate” or “severe” based on the severity criteria defined by the European Crohn’s and Colitis Organisation (ECCO) [7]. Moderate CD was considered with biologic therapy and severe CD with biologic therapy and a history of peri-anal disease and/or IBD-related surgery. Disease activity was determined by clinical examination, the results of the last colonoscopy or MRI and the last fecal calprotectin level. The patients were in clinical activity with a Harvey Bradshaw (HB) score ≥4 and in clinical remission with HB <4, and in deep remission with HB score <4 and lack of activity on MRI and/or Simple Endoscopic Score for CD <3 on total ileocolonoscopy and/or fecal calprotectin level <250 mcg/g [24].

The following data were recorded: age, sex, IBD characteristics (disease duration, Montreal classification [25]), prior IBD surgery, current biologic treatment, concomitant immunomodulatory treatment (azathioprine, 6-mercaptopurine or methotrexate), corticosteroids use and patient comorbidities, body mass index (BMI), systolic and diastolic blood pressure, dyslipidemia and family cardiovascular history. Weight and height were collected from the medical records to calculate BMI (weight/height^2^ in kg/m^2^). Blood pressure was measured by using the Omron M2 (Kyoto, Kansai, Japan) device and calculated as the mean of 3 measurements after at least a 5 min sitting position. Dyslipidemia was defined by low density cholesterol level >4.1 mmol/L and/or triglycerides level >1.7 mmoL/L and/or cholesterol-lowering treatment and family cardiovascular history by myocardial infarction (age < 55 years in a man, <65 years in a woman) or stroke (age < 45 years) in a first-degree relative.

### 2.5. Statistical Analysis

Continuous variables are described with mean (standard deviation (SD)) and categorical variables with number (percentage); any missing data are indicated. Comparisons between the controls and the CD patients involved the Mann–Whitney test for continuous variables and the chi-square test (or Fisher’s exact test) for categorical variables.

To identify differences in the OCT-A variables between the 3 groups (controls, moderate and severe CD), an age- and sex-adjusted analysis of covariance (ANCOVA) was used, followed by post-hoc pairwise comparisons for the significant variable. The post-hoc pairwise comparisons were corrected for multiple testing by the Benjamini–Hochberg method to minimize type I error [26]. ANCOVA tests were adjusted for age and sex in complement to the age- and sex-matching to minimize their residual confounding effect [27], because they can affect OCT-A variables [28]. Additional ANCOVA was performed to test the association between the disease activity and the main OCT-A variables. The SCP-FI and the DCP-FI showed a skewed distribution of residues and required a natural logarithmic (log) transformation.

To assess the ability of OCT-A to stage CD severity, the area under the receiver operating characteristic curve (AUC) values were ascertained for significant variables by using logistic regression. Similarly, multivariate AUC analysis was performed and, based on the significant variables who were not correlated to each other. Two-tailed *p* < 0.05 was considered statistically significant. Statistical analyses were performed with R 4.0.3 (Vienna, Austria).

## 3. Results

Among the 157 eligible participants, 148 eyes of 148 participants were analyzed (74 CD patients and 74 controls). Four participants with CD and five controls were excluded because of low-quality images (Q-score <6 or artefacts) for both eyes.

The demographic and clinical characteristics of the population are described in Table 1. We compared 33 patients with moderate CD (14 men (42%); mean (SD) age, 43 (16) years) and 41 with severe CD (19 men (46%); mean (SD) age, 46 (12) years) with 74 controls. The three groups (controls, moderate CD and severe CD) did not differ in potential confounding factors (smoking, dyslipidemia, systolic/diastolic blood pressure, BMI, personal cardiovascular history and Q-score; all *p* ≥ 0.20).

Disease characteristics in CD patients are described in Table 2.

The two CD groups did not differ in sex (*p* = 0.74), age (*p* = 0.39), CD location (*p* = 0.24) or remission (clinical, *p* = 0.61; deep, *p* = 0.47). Severe CD patients showed more penetrating damage (B3) (*p* = 0.024) and moderate CD patients less non-penetrating non-stricturing damage (B1) (*p* = 0.002). More than half of the severe CD patients had peri-anal disease (57%) and almost three quarters had an IBD-related surgery (73%). The mean (SD) disease duration was higher with severe than moderate CD (17 (10) vs. 11 (9) years, *p* = 0.013). Table 3 shows the associations of CD severity with microvasculature and structural retinal variables after adjustment for age and sex.

The three analyzed groups significantly differed in the FAZ area (*p* = 0.001), ln (SCP-FI) (*p* = 0.048), whole and parafoveal SCP-VD (both *p* < 0.05) and RPC-VD (*p* = 0.035), with no difference in DCP-VD or DCP-FI. Figure 3 illustrates the significant associations between the macular microvascular variables and the disease status.

Compared with the controls, the severe CD patients showed reduced mean whole and parafoveal SCP-VD (both *p* < 0.05), reduced ln (SCP-FI) (although not significant (*p* = 0.06)) and increased mean FAZ area (*p* = 0.001). Compared with the moderate CD patients, the severe CD patients showed an increased mean FAZ area (*p* = 0.010) and RPC-VD (*p* = 0.034) and a decreased mean SCP-FI (although not significant (*p* = 0.08)). The microvasculature did not differ between the controls and the moderate CD patients. The OCT-A variables did not differ between patients with clinical or deep remission (Appendix A).

The three groups did not differ in mean RNFL and GCC. Severe CD patients showed a significant retinal thinning compared with the controls (*p* < 0.001).

Figure 4 and Table 3 present the results for the AUC analysis for significant variables and the overall AUC. The AUC values ranged from 0.59 to 0.69: RPC-VD, 0.59; ln (SCP-FI), 0.59; parafoveal SCP-VD, 0.59; whole SCP-VD, 0.60; parafoveal RT, 0.65; FAZ area, 0.69; general AUC, 0.75.

## 4. Discussion

In this cross-sectional study, we observed the differences in retinal microcirculation among patients with moderate CD (under biologic therapy) and severe CD (under biologic therapy and history of perianal disease and/or IBD-related surgery) and age- and sex-matched controls. The severe CD patients showed reduced SCP-VD, SCP-FI, parafoveal retinal thickness and RPC-VD and increased FAZ area compared with the controls, and increased FAZ area and RPC-VD compared with the moderate CD patients. These findings suggest objective changes in retinal microcirculation in patients with severe CD.

The severe CD patients showed a rarefaction of the macular retinal microvascular network as a consequence of a microvascular interruption. We found lower whole and parafoveal SCP-VD, lower SCP-FI and larger FAZ area in the patients with severe CD than the controls. The mechanisms of retinal microvascular changes in CD patients have never been studied. An enlargement of the FAZ has been described in pathologies causing retinal capillary ischemia such as diabetic retinopathy [29] and lupus erythematosus [20]. The microcirculatory damage observed in our population of CD patients may be explained by the inflammation and arthrosclerosis and prothrombotic state that the disease induces. CD is associated with an increased risk of venous or arterial or venous events: the risk of venous thromboembolism is higher in IBD patients than controls, especially in disease flare (hazard ratio 8.4; 95% confidence interval (CI) 5.5–12.8) [30]; coronary artery disease is four times higher than in matched controls [31] and cerebrovascular events are more likely, especially in young patients (incident rate ratio 19.9; 95% CI, 1.8–219) [32]. This prothrombotic state is possibly related to the increase in inflammatory cytokine levels (high sensitivity C-reactive protein (CRP), IL-6 and TNF-α) [33], which are proatherogenic [34], decrease endothelial progenitor cell survival [35] and cause endothelial dysfunction [36]. An elevated CRP level is associated with increased arterial thrombotic risk [37] and the persistence of residual inflammation in patients with CD may be associated with increased risk of arterial events [38]. This finding could explain the degradation of retinal microcirculation in patients with severe CD, in whom levels of proinflammatory cytokines were probably higher and more persistent than in other states [10]. Decreased SCP has also been associated with increased cardiovascular risk in patients hospitalized for acute coronary syndrome [14].

The participants in our study had no personal cardiovascular history, and they could be followed over time to determine whether cardiovascular events occur more frequently in the group with microcirculatory involvement. OCT-A findings could then be a predictor of cardiovascular risk and help in the management and prevention of cardiovascular risk in these patients, which is a real issue [39].

Some studies highlight the potential of anti-TNF-α agents (biologics) to reverse pre-existing subclinical atherosclerosis by reducing chronic inflammation [40,41]. OCT-A could help answer this question in a study comparing retinal density before and after the introduction of biologics. Biological therapies could, in the short term, improve retinal flow by decreasing CRP-associated blood viscosity and, in the long term, limit the decrease in retinal microvascular density by limiting arthrosclerosis and prothrombotic risks.

Nakayama et al., described an increase in the FAZ area in patients with active chronic IBD (CD and ulcerative colitis, *n* = 72) [16]. We did not find any differences in microvascular variables by CD activity after adjusting for potential confounders. The univariate analysis of Nakayama et al., did not consider potential confounders, which thus limited the validity of this result. However, drawing firm conclusions from this study is difficult, because it included a different population from ours (CD patients under biologic therapy vs. CD patients and patients with ulcerative colitis). In general, our study and the Nakayama et al., study were cross-sectional and analyzed the activity at a certain time and not over time. The altered VD or FAZ area could be due more to chronic inflammation-related capillary rarefaction rather than acute inflammatory episodes, therefore the number of active trimesters would have been a more relevant factor to associate with microvascular density.

We did not observe changes in FI in active CD. This finding could be explained by the use of biologics (mostly anti-TNF-α agents) in our CD patients. Indeed, Bonnin et al., visualized a normalization of the retrobulbar flow in doppler ultrasound after infliximab in active CD patients, due to a decrease in fibrinogen, which reduces blood viscosity [42]. Moreover, the retinal flow might be sensitive to acute inflammatory events, because CRP reduces the vasomotor effect of nitric oxide on the endothelium [43]. In brief, our results cannot exclude a reduction in retinal flux in active CD under biologic therapy.

The use of OCT-A findings as a potential biomarker of severity has never been studied in CD. Our AUC analyses of individual retinal factors were not sufficiently discriminating (AUC from 0.60–0.69) for them to be independent biomarkers of disease with cutoffs that define CD severity. However, the AUC value of 0.75, combining three microvascular factors, was good. This finding suggests that, combined with severity criteria defined by the ECCO guidelines [7], this retinal non-invasive biomarker could help clinicians assess the severity of CD.

Contrary to the SCP, macular DCP was not altered and mean RPC-VD was lower in severe than moderate CD. DCP and RPC do not seem primarily affected in CD. RPC-VD may increase to compensate for a higher oxygen demand secondary to altered SCP, because SCP and RPC are interconnected [44]. This hypothesis of microvascular adaptation is supported by the study of Kasumovic et al., who described a paradoxical increase in RPC-VD in the most severe stages of chronic kidney disease [45].

Mean GCC and RNFL did not differ among the groups, and parafoveal RT was decreased in severe CD. In fact, parafoveal RT is associated with enlargement of the FAZ area because the two factors were highly correlated (Pearson correlation, *r* = −0.78, *p* < 0.001). A longitudinal follow-up is needed to determine whether parafoveolar retinal thinning is secondary to enlarged FAZ.

### Limitations and Strengths

The first limitation is the lack of a recognized classification of CD severity in the literature, which may have introduced bias in our study; however, it was reduced by the use of relevant criteria defined by the ECCO guidelines [7] to classify the disease severity. The lower proportions of fistulizing disease (B3) (15% vs. 39%, *p* = 0.024) and disease duration (11 vs. 17 years, *p* = 0.013) in moderate CD leads us to believe that the classification was well performed. Second, this study involved highly selected CD patients (on biologic therapy) and results cannot be extrapolated to less severe CD. Third, we analyzed 14 retinal factors and did not decrease the significance level to maintain the trend of the results. This situation may have increased the type I error rate, therefore significant results should be interpreted with this perspective. Fourth, the limited sample size did not allow us to demonstrate differences in macular microvascularization between the controls and the moderate CD patients. As an illustration, the number of subjects needed to objectify a difference in SCP-VD parafoveal between these two subgroups was 584 compared with our 107 moderate or control subjects (post hoc sample size calculation, 80% power). Finally, the cross-sectional nature of our study did not allow us to ascertain cause.

The first strength is that the data collection was systematized, which limits the risk of missing data in the final analysis. This is the first study to describe retinal microcirculatory changes in CD patients by severity. It involved a relatively young population, which allowed for high-quality images for almost all eligible patients, thus limiting measurement bias. The use of only automatic analyses also limited this bias. We also performed AUC analyses with a multivariate AUC analysis, which suggests that OCT-A combined with ECCO criteria could help clinicians better assess the disease severity. Mostly, we did not find differences in potential confounders between groups, and we limited the bias by matching for age and sex.

## 5. Conclusions

We observed altered retinal microvascularization in severe versus moderate CD and age- and sex-matched controls. The severe CD patients had reduced VD and a larger FAZ area compared with the control participants. The FAZ area and the RPC-VD was significantly different between severe and moderate CD. These results need to be confirmed in larger studies, but retinal microcirculation could be a promising new biomarker of CD severity.

## Figures and Tables

**Figure 1 jpm-12-00230-f001:**
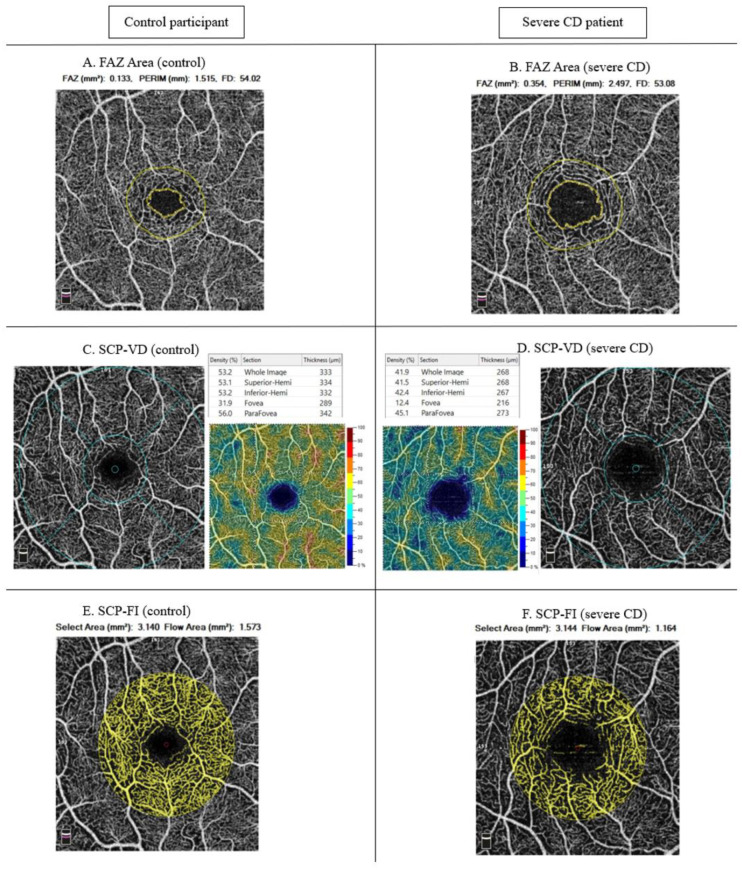
Examples of OCT-A images for controls and patients with severe CD. Abbreviations: CD—Crohn’s disease; FAZ—foveal avascular zone; FI—flow index; SCP—superficial capillary plexus; VD—vascular density.

**Figure 2 jpm-12-00230-f002:**
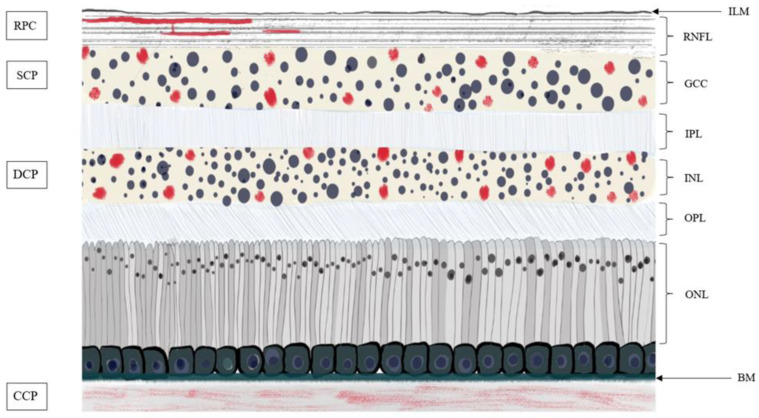
Drawing of vascular capillary plexus in retina. The capillaries are represented by red dots and the cell nuclei by dark blue dots. The RPC is arranged parallel to the nerve fibers. Abbreviations: BM—Bruch’s membrane; CCP—choriocapillaris plexus; DCP—deep capillary plexus; GCC—ganglion cell complex; ILM—inner limiting membrane; INL—inner nucleus layer; IPL—inner plexiform layer; ONL—outer nucleus layer; OPL—outer plexiform layer; RNFL—retinal nerve fiber layer; RPC—radial peripapillary capillaries; SCP–superficial capillary plexus.

**Figure 3 jpm-12-00230-f003:**
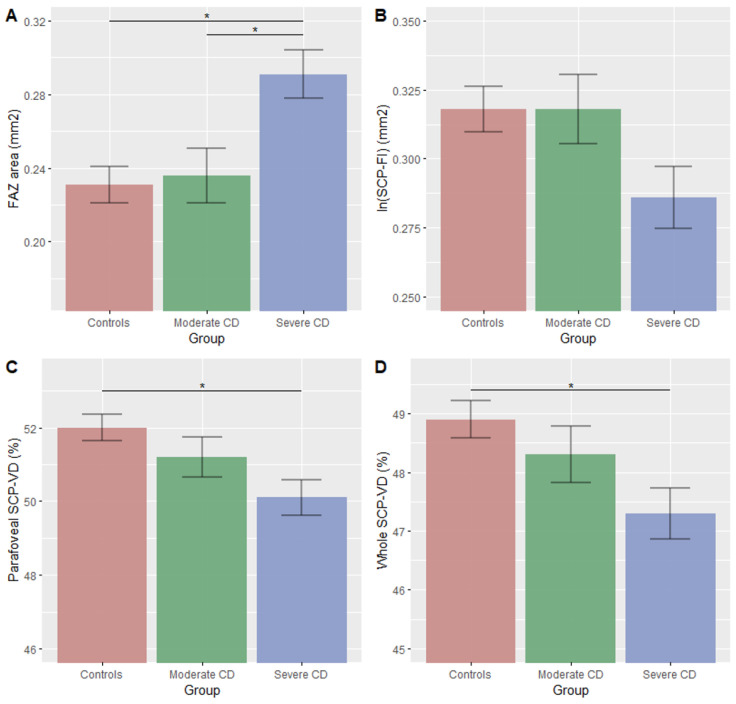
Age- and sex-adjusted ANCOVA, significant microvascular macular features according to Crohn’s disease severity. (**A**) FAZ area, *p*-value post-hoc ANCOVA: controls vs. moderate CD, 0.01; controls vs. severe CD, 0.001; (**B**) ln (SCP-FI), *p*-value post-hoc ANCOVA: controls vs. moderate CD, 0.08; moderate vs. severe CD, 0.08; (**C**) Parafoveal SCP-VD, *p*-value post-hoc ANCOVA: controls vs. severe CD, 0.009; (**D**) Whole SCP-VD, *p*-value post-hoc ANCOVA: controls vs. severe CD, 0.006. Data are mean (SE) according to CD status adjusted for age and sex. Significant *p*-value post-hoc ANCOVA (<0.05) are represented by *. Abbreviations: CCP—choriocapillaris plexus; DCP—deep capillary plexus; FAZ—foveal avascular zone; FI—flow index; SCP—superficial capillary plexus; VD—vascular density.

**Figure 4 jpm-12-00230-f004:**
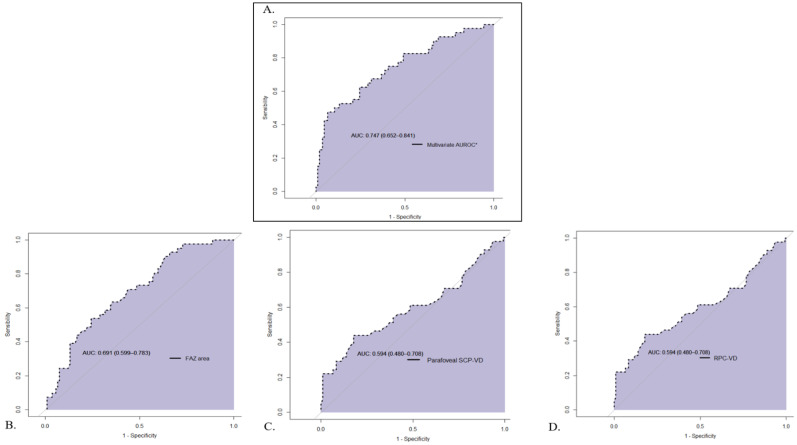
Ability of OCT-A to stage Crohn’s disease severity, area under the receiver operating characteristic curve (AUC). Area under the receiver operating characteristic curve (AUC) values were ascertained for significant microvascular features by using logistic regression, with diagnosis of severe CD as the outcome variable. The multivariate AUC and the AUC values for the independent factors used for the multivariate AUC analysis are plotted. (**A**) Multivariate AUC (based on the non-correlated and significant parameters; i.e., FAZ, RPC-VD and parafoveal SCP-VD); (**B**) FAZ area AUC; (**C**) Parafoveal SCP-VD AUC; (**D**) RPC-VD AUC. Abbreviations: AUC—area under the receiver operating characteristic curve; FAZ—foveal avascular zone; RPC—radial peripapillary capillaries; SCP—superficial capillary plexus; VD—vascular density.

**Table 1 jpm-12-00230-t001:** Characteristics of the study population with and without Crohn’s disease.

	Moderate CD, N = 33	Severe CD, N = 41	*p*-Value ^a^
Age (years)	43 (16)	46 (12)	0.39
Sex (male), n (%)	14 (42)	19 (46)	0.74
Age at diagnosis (years)	32 (15)	29 (12)	0.51
Duration of disease (years)	11 (9)	17 (10)	0.013
CD behavior, n (%) ^b^			
Non-penetrating non-stricturing (B1)	18 (55)	8 (20)	0.002
Stricturing (B2)	7 (21)	14 (34)	0.22
Penetrating (B3)	5 (15)	16 (39)	0.024
CD location, n (%) ^c^			0.24
Colonic (L2)	6 (18)	10 (25)	
Ileal (L1)	18 (55)	14 (35)	
Ileocolonic (L3)	9 (27)	16 (40)	
Isolated upper disease (L4)	0 (0)	0 (0)	
Peri-anal disease, n (%)	0 (0)	22 (54)	<0.001
Past IBD-related surgery, n (%)	0 (0)	30 (73)	<0.001
Clinical remission, n (%)	25 (76)	28 (68)	0.61
Deep remission, n (%)	17 (55)	17 (46)	0.47
Missing value	2	4	
Type of biologic, n (%)			0.088
Infliximab (anti TNF-α)	24 (73)	19 (46)	
Adalimumab (anti TNF-α)	3 (9.1)	3 (7.3)	
Guselkumab (anti IL-23)	0 (0)	1 (2.4)	
Ustekinumab (anti IL-12/IL-23)	3 (9.1)	7 (17)	
Vedolizumab (anti integrin α4-β7)	1 (3.0)	8 (20)	
Filgotinib (anti JAK)	1 (3.0)	0 (0)	
Rizankizumab (anti IL-23)	1 (3.0)	3 (7.3)	
Corticosteroid use, n (%)	2 (6.1)	1 (2.4)	0.58
Concomitant combination therapy, n (%)	3 (9.1)	6 (15)	0.72

Data are mean (SD) unless indicated. ^a^ Wilcoxon rank sum exact test; Wilcoxon rank sum test; chi-square test; Fisher’s exact test; ^b^ a *p* value is associated with each modality because it is possible to be both B2 and B3; ^c^ one patient had isolated peri-anal CD.

**Table 2 jpm-12-00230-t002:** Characteristics of patients with Crohn’s disease.

Potential Confounders	Controls, N = 74	Moderate CD, N = 33	Severe CD, N = 41	*p*-Value ^a^
Age (years)	44 (14)	43 (16)	46 (12)	0.68
Sex (male), n (%)	33 (45)	14 (42)	19 (46)	0.94
Tobacco use, n (%)				0.85
Cessation	10 (14)	6 (18)	6 (15)	
Current	18 (24)	9 (27)	13 (32)	
No	46 (62)	18 (55)	22 (54)	
Dyslipidemia history, n (%)	2 (2.7)	1 (3.0)	2 (4.9)	0.84
Family cardiovascular history, n (%)	8 (11)	7 (21)	9 (22)	0.20
BP: systolic/diastolic (mmHg)	121 (19)/77 (9)	123 (16)/77 (13)	122 (17)/77 (10)	0.81/0.88
BMI (kg/m^2^)	23.6 (3.3)	23.4 (3.7)	24.0 (4.2)	0.82
SE (Diopters)	−0.08 (1.60)	0.56 (1.58)	−0.05 (1.23)	0.20
Missing value	0	1	1	
IOP (mmHg)	14.18 (2.98)	14.71 (2.31)	14.31 (3.16)	0.47
Missing value	0	2	2	
Q-score	8.3 (0.9)	8.3 (0.9)	8.2 (0.9)	≥0.99

Data are mean (SD) unless indicated. ^a^ Kruskal–Wallis rank sum test; Pearson’s chi-squared test; Fisher’s exact test. Abbreviations: BP—blood pressure; BMI—body mass index; CD—Crohn’s disease; IOP—intra-ocular pressure; Q-score—quality image index; SE—spherical equivalent.

**Table 3 jpm-12-00230-t003:** Age- and sex-adjusted analysis of covariance, association of retinal variables with Crohn’s disease severity.

	Controls (1), N = 74	Moderate CD (2), N = 33	Severe CD (3), N = 41	Global P ^a^	BH P ^b^	AUC ^c^
Microvasculature parameters						0.75
FAZ area (mm^2^)	0.231 (0.01)	0.236 (0.01)	0.291 (0.01)	0.001	1–3: 0.001 2–3: 0.010	0.69
Macular SCP-VD (%)						
Whole	48.9 (0.3)	48.3 (0.5)	47.3 (0.4)	0.012	1–3: 0.009	0.60
Parafoveal	52.0 (0.4)	51.2 (0.5)	50.1 (0.5)	0.008	1–3: 0.006	0.59
Macular DCP-VD (%)						
Whole	54.0 (0.4)	54.0 (0.6)	53.5 (0.5)	0.67	-	
Parafoveal	55.9 (0.4)	55.6 (0.6)	55.3 (0.5)	0.71	-	
ONH VD (%)						
Whole image	49.6 (0.3)	49.1 (0.4)	50.0 (0.4)	0.22	-	
Inside	51.1 (0.5)	51.4 (0.8)	51.3 (0.7)	0.84	-	
RPC	51.7 (0.3)	50.5 (0.5)	52.2 (0.4)	0.035	1–2: 0.067 2–3: 0.034	0.59
Macular FI (mm^2^)						
ln (SCP-FI)	0.318 (0.008)	0.318 (0.01)	0.286 (0.01)	0.048	1–3: 0.06 2–3: 0.08	0.59
ln (DCP-FI)	−0.97 (0.06)	−0.90 (0.09)	−0.99 (0.09)	0.075	-	
CCP-FI	2.13 (0.014)	2.11 (0.2)	2.11 (0.02)	0.74	-	
OCT features (μm)						
Parafoveal RT	323 (1.9)	314 (2.9)	309 (2.6)	<0.001	1–3: <0.001	0.65
Mean GCC	99.2 (0.7)	97.2 (1.1)	97.5 (1.0)	0.23	-	
Mean RNFL thickness	103 (1.1)	101 (1.6)	103 (1.4)	0.79	-	

Data are mean (SEM). ^a^ Model adjusted for sex and age: global *p* value of a difference among the 3 groups. ^b^ Adjusted *p* values reported with the Benjamini–Hochberg method, calculated if the global *p*-value was <0.05. ^c^ Multivariate logistic regression was performed to obtain this AUC. The absence of multicollinearity was verified by using the variance inflation factor; the remaining explanatory factors were FAZ, RPC-VD and VD parafoveal. Abbreviations: AUC—area under the receiver operating characteristic curve; BH—Benjamini–Hochberg; CCP—choriocapillaris plexus; DCP—deep capillary plexus; FAZ—foveal avascular zone; FI—flow index; GCC—ganglion cell complex; ONH—optic nerve head; RNFL—retinal nerve fiber layer; RPC—radial peripapillary capillaries; RT—retinal thickness; SCP—superficial capillary plexus; SEM—standard error of the mean; VD—vascular density.

## Data Availability

The data presented in this study are available on request from the corresponding author.

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
