# Peer review of "Retinal Microcirculation Changes in Crohn’s Disease Patients under Biologics, a Potential Biomarker of Severity: A Pilot Study"

_jpm, 2022, doi:10.3390/jpm12020230_

Round 1
Reviewer 1 Report
the study is interesting, there are aspects that might be clarified:
- if there is a intra-vessels thromb formation risk due to CD how this interact with retinal microcirculation?
- did the AA recorded in the whole group any vessel interruption?
- there are any correlations with blood pressure in CD group?
- in which way biological therapy interact with microcirculation damage?
Author Response
POINT-BY-POINT RESPONSE FORM
Manuscript #: jpm-1545288
Manuscript title: Retinal microcirculation changes in Crohn’s disease patients under biologics: a Potential Biomarker of Severity
We sincerely thank the reviewer for his help in the improvement and clarification of this paper. We incorporated all comments in the manuscript. All changes were highlighted in yellow.
|
Suggestion, Question, or Comment from the Reviewer#1 |
Author’s Response |
Change in the Manuscript |
|
If there is a intra-vessels thromb formation risk due to CD how this interact with retinal microcirculation?
|
We thank the reviewer for this clarification about the interaction between intravascular thrombus formation in Crohn's disease and its impact on the retinal microcirculation |
We added in the introduction, lines 61-65: “In the same way, altered OCT-A findings were described in rheumatoid arthritis, another chronic systemic inflammatory disease with increased cardiovascular risk [16,17], and in lupus erythematosus which is also associated with a high thrombotic risk [18]. The increased cardiovascular and prothrombotic risk in CD could be associated with a rarefaction of the retinal microcirculation, which can be observed in OCT-A by a decrease of the microvascular density or an enlargement of the central avascular zone[19].” |
|
Did the AA recorded in the whole group any vessel interruption?
|
We thank the reviewer for giving us the opportunity to clarify our results. The interruption of microvessels is demonstrated in OCT-A by a decrease of the vascular density and by a widening of the central avascular zone. Vessel interruption cannot be visualized by qualitative analysis of OCT-A "en face” image except if there is an arterial or arteriolar branch occlusion (which is not part of our study where the patients were free of retinal disease). |
We added further explanation of the results in the discussion, lines 251-252: “Severe CD patients showed a rarefaction of the macular retinal microvascular network, as a consequence of a microvascular interruption.”
|
|
There are any correlations with blood pressure in CD group? |
We thank the reviewer for giving us the opportunity to specify how blood pressure was taken into account in the statistical analysis to verify that it was not a confounding factor.
There was no statistically significant difference between moderate and severe CD patients for systolic (123mmHg vs. 122mmHg, p=0.81, Kruskal-Wallis) and diastolic blood pressure (77 mmHg vs. 77 mmHg, p=0.88, Kruskal-Wallis). The ANCOVA model therefore did not have to be adjusted for these covariates, which were therefore not correlated with the differences in OCT-A parameters between the groups. |
|
|
In which way biological therapy interact with microcirculation damage?
|
We thank the reviewer for giving us the possibility to clarify the effects of immunomodulators on the retinal microcirculation.
To our knowledge, no study has evaluated the effect of biological therapy on retinal microcirculation. |
We added a sentence in the discussion, lines 280-282: “Biological therapies could in the short term improve retinal flow by decreasing CRP-associated blood viscosity, and in the long term limit the decrease in retinal microvascular density by limiting arthrosclerosis and prothrombotic risk.” |
Reviewer 2 Report
The paper is interesting, and the methodology is sound; the case and controls were age and gender-matched. The use of statistical methods is adequate and well precise. The plagiarism check did not reveal a significant overlap with published data. The authors corrected for multiple comparisons.
The main drawback of the study is the small sample size that weakens the observed results. To better explain this, the authors are encouraged to calculate the sample size needed for an adequate power (80%) with their primary outcomes. This idea needs to be highlighted in the limitations section.
Among the 157 eligible participants, 148 eyes were analyzed.. why were nine others omitted? Were they outliers? The authors need to explain this issue.
Some sentences across the text, including the introduction, lack reference(s).. The authors are encouraged to add them..
Author Response
Manuscript #: jpm-1545288
Manuscript title: Retinal microcirculation changes in Crohn’s disease patients under biologics: a Potential Biomarker of Severity
We sincerely thank the reviewer for his help in the improvement and clarification of this paper. We incorporated all comments in the manuscript. All changes were highlighted in yellow.
|
Suggestion, Question, or Comment from the Reviewer#1 |
Author’s Response |
Change in the Manuscript |
|
The main drawback of the study is the small sample size that weakens the observed results. To better explain this, the authors are encouraged to calculate the sample size needed for an adequate power (80%) with their primary outcomes. This idea needs to be highlighted in the limitations section.
|
We thank the reviewer for pointing this out.
To illustrate the limited number of subjects to demonstrate differences between moderate CD patients and controls, we calculated the number of subjects needed to visualize a difference in one of the macular microvascular parameters between these two subgroups. |
We added in the limitations section, lines 329-333: “Fourth, the limited sample size did not allow us to demonstrate differences in macular microvascularisation between controls and moderate CD patients. As an illustration, the number of subjects needed to objectify a difference in SCP-VD parafoveal between these 2 subgroups was 584 compared with our 107 moderate or control subjects (post hoc sample size calculation, 80% power).”
|
|
Among the 157 eligible participants, 148 eyes were analyzed. Why were nine others omitted? Were they outliers? The authors need to explain this issue.
|
We thank the reviewer for giving us the opportunity to clarify this item.
Among the 157 eligible participants, 9 participants (5 controls and 4 participants with CD) were excluded because of low quality image (line 86: exclusion criteria “quality index [Q-score] <6 or artefacts”) |
We added in results, line 173: “Among the 157 eligible participants, 148 eyes of 148 participants were analyzed (74 CD patients and 74 controls). Four participants with CD and 5 controls were excluded because of low-quality images (Q-score <6 or artefacts) for both eyes.
|
|
Some sentences across the text, including the introduction, lack reference(s).. The authors are encouraged to add them.. |
We thank the reviewer for giving us the opportunity to add some references |
We add the references below: Line 40: “It is a systemic inflammatory disease, characterized by an inappropriate inflammatory response to modified gut microbiota in patients with genetic susceptibility [2].”
Line 50: The therapeutic target is to achieve clinical and endoscopic remission to avoid complications and recourse to surgery [7].
Line 140: “The following data were recorded: age, sex, IBD characteristics (disease duration, Montreal classification [25]),..”
|

Round 2
Reviewer 1 Report
may be interesting discuss other causes in which there is an enlargement of FAZ. but also in this version the paper sounds good.
Author Response
Manuscript #: jpm-1545288
Manuscript title: Retinal microcirculation changes in Crohn’s disease patients under biologics, a Potential Biomarker of Severity: a pilot study
We sincerely thank the reviewer for his help in the improvement and clarification of this paper. We incorporated all comments in the manuscript. All changes were highlighted in yellow.
|
Suggestion, Question, or Comment from the Reviewer#1 |
Author’s Response |
Change in the Manuscript |
|
It may be interesting discuss other causes in which there is an enlargement of FAZ. but also in this version the paper sounds good. |
We thank the reviewer for this suggestion. |
We added in the discussion, lines 256-257: “An enlargement of the FAZ has been described in pathologies causing retinal capillary ischemia such as diabetic retinopathy[29] and lupus erythematosus[20]” |
ur comments, attached is a word document of our response
